# Towards Cross Domain Generalization of Hamiltonian Representation via Meta Learning

**Yeongwoo Song[1]    Hawoong Jeong[1][2]***
[1]Department of Physics, KAIST    [2]Center for Complex Systems, KAIST
`ywsong1025@kaist.ac.kr, hjeong@kaist.edu`

## Abstract

Recent advances in deep learning for physics have focused on discovering shared representations of target systems by incorporating physics priors or inductive biases into neural networks. While effective, these methods are limited to the system domain, where the type of system remains consistent and thus cannot ensure the adaptation to new, or unseen physical systems governed by different laws. For instance, a neural network trained on a mass-spring system cannot guarantee accurate predictions for the behavior of a two-body system or any other system with different physical laws. In this work, we take a significant leap forward by targeting cross domain generalization within the field of Hamiltonian dynamics. We model our system with a graph neural network (GNN) and employ a meta learning algorithm to enable the model to gain experience over a distribution of systems and make it adapt to new physics. Our approach aims to learn a unified Hamiltonian representation that is generalizable across multiple system domains, thereby overcoming the limitations of system-specific models. We demonstrate that the meta-trained model captures the generalized Hamiltonian representation that is consistent across different physical domains. Overall, through the use of meta learning, we offer a framework that achieves cross domain generalization, providing a step towards a unified model for understanding a wide array of dynamical systems via deep learning.

## 1 Introduction

Deep learning has succeeded in many application areas such as image classification, image generation, natural language processing, and so on (Reed et al., 2016; Tan & Le, 2019; Raffel et al., 2020; Gu et al., 2022). One of the major roles of such accomplishment was capable of parameterizing useful representations from data with neural networks (Chen et al., 2020; Van Den Oord et al., 2017; Hamilton et al., 2017). However, grafting deep learning to physics is yet another problem. They struggle to learn conservation laws or implicit physical geometries or symmetries. Although various studies make the model learn the conservative quantities or symmetries inside the system (Greydanus et al., 2019; Sanchez-Gonzalez et al., 2019; Liu & Tegmark, 2021), these approaches are system-specific, which means that if a model is trained for one system, it cannot easily adapt to another system with different physical laws. Since systems whose physics is unknown have more sparse data, these flaws make the standard supervised learning methods harder to learn the physics of the system.

One of the current extents of this limitation is crafting the generalization of a fixed type of system across various parameter environments (Lee et al., 2021; Yin et al., 2021; Kirchmeyer et al., 2022; Li et al., 2023). Here, one can think of an extension of building a unified model that is not limited to adapting to various parameter environments within a single system type, but instead generalizes across the vast landscape of different system types. In this paper, we focus our objectives on systems that Hamiltonian mechanics can formulate. Systems governed by Hamiltonian mechanics inhere symmetries, and their state trajectories are laid on the symplectic manifold (Libermann & Marle, 2012; Arnol'd, 2013). Moreover, almost all of the physics in our nature has its own conservation laws, and Hamiltonian mechanics relates the state of the system to its corresponding conservative quantities (usually energy).

---

*Correspondence to Hawoong Jeong.

In this work, we search for the generalized representation that shares the essence of Hamiltonian mechanics that a neural network can learn. In doing so, we aim to make the model easily adaptable to unseen physics.

Our contributions are summarized as follows.

- Utilizing the meta learning algorithm, we extend its application to the data distribution across the Hamiltonian manifold, encompassing various types of physical systems. This enables our model to generalize effectively to system data with unknown physics, enhancing its versatility.

- We begin by evaluating the model's performance on simpler examples, such as the mass-spring and pendulum systems. Subsequently, we expand our evaluation to more complex scenarios characterized by chaotic behavior, such as the Hénon-Heiles and magnetic-mirror systems. This demonstrates the robustness and applicability of our approach across a wide range of dynamical systems.

- To gain insights into the superior performance of the meta-trained model compared to other baselines, we conduct an in-depth analysis of the learned representations within the neural network. This investigation provides a deeper understanding of the underlying factors contributing to the model's effectiveness due to meta learning.

- We use the term *domain* to refer to different types of physical systems governed by distinct laws. To the best of our knowledge, we are the first to explore cross-domain generalization across diverse dynamical system types enabling easy adaptation to new physics, surpassing prior studies limited to fixed system types.

## 2 PRELIMINARIES

### 2.1 HAMILTONIAN MECHANICS

Hamiltonian mechanics is one of the concrete frameworks to describe the state of a dynamical system. It begins by defining a vector composed of canonical coordinates $\boldsymbol{x} = (\boldsymbol{q}, \boldsymbol{p})$. Here $\boldsymbol{q}$ and $\boldsymbol{p}$ denotes the generalized coordinate and its canonical momentum respectively; $\boldsymbol{q} = (q_1, q_2, \ldots, q_N)$, $\boldsymbol{p} = (p_1, p_2, \ldots, p_N)$, where $N$ is the degree of freedom of the given system. Then, Hamiltonian ($\mathcal{H}$) is defined such that Hamilton's equation (Equation 1) is held. Usually, the Hamiltonian of the system corresponds to its energy where it is to be conserved.

$$\frac{d\boldsymbol{q}}{dt} = \frac{\partial \mathcal{H}}{\partial \boldsymbol{p}}, \ \frac{d\boldsymbol{p}}{dt} = -\frac{\partial \mathcal{H}}{\partial \boldsymbol{q}} \tag{1}$$

Here, the Hamiltonian vector field $\boldsymbol{X}_{\mathcal{H}} = \left( \frac{\partial \mathcal{H}}{\partial \boldsymbol{p}}, -\frac{\partial \mathcal{H}}{\partial \boldsymbol{q}} \right) = J\nabla_{\boldsymbol{x}}\mathcal{H}_{\theta}$, where $J = \begin{bmatrix} 0 & I_n \\ -I_n & 0 \end{bmatrix}$, is known as symplectic, which allows the Hamiltonian to lie on a symplectic manifold. Symplectic manifold is one of the essences of differential geometry, which naturally arises from the formulations of Hamiltonian mechanics (Libermann & Marle, 2012; Arnol'd, 2013). It provides the generalization of the phase space of a closed system, allowing one to probe the time evolution of the system.

### 2.2 LEARNING PHYSICAL DYNAMICS FROM NEURAL NETWORKS

Various works use neural networks to analyze the time evolution, or the conservation law of the physical system (Greydanus et al., 2019; Sanchez-Gonzalez et al., 2019; Cranmer et al., 2020; Kasim & Lim, 2022; Greydanus et al., 2023). For example, Hamiltonian Neural Network (HNN) (Greydanus et al., 2019) parameterizes the Hamiltonian of the system with a neural network. HNN learns the dynamics of the given system by inducing a physical bias to the loss function. Making use of Equation 1, HNN predicts the dynamics of the system incorporating the symplectic gradient inside the loss function as in Equation 2.

$$\mathcal{L}_{\text{HNN}} = \left\| \frac{\partial \mathcal{H}_{\theta}}{\partial \boldsymbol{p}} - \frac{\partial \boldsymbol{q}}{\partial t} \right\|_2 + \left\| \frac{\partial \mathcal{H}_{\theta}}{\partial \boldsymbol{q}} + \frac{\partial \boldsymbol{p}}{\partial t} \right\|_2 \tag{2}$$

As an alternative way to encode the physical bias into the objective function, the Variational Integrator Network (VIN) (Saemundsson et al., 2020) has been introduced, which preserves the geometric structure of physical systems. The architecture of VIN is designed to fit the discrete-time equations of motion of the given dynamical system. Thus, it provides interpretability and more efficient learning by preserving the manifold geometry inherent in physical systems directly in the model architecture.

While the theoretical understanding of inductive biases that underlie the advantages of neural networks remains elusive, recent studies attempt to shed light on the influence of such specific biases. For example, one demonstrates that contrary to conventional wisdom, the performance improvement in HNNs arises from the second-order structure, rather than the symplectic bias (Gruver et al., 2022). Nevertheless, a number of works focus on leveraging the inherent nature of physical systems, such as their symplectic structure and the principle of least action, to enhance the predictive capabilities of neural networks for system dynamics (Chen et al., 2021; Greydanus et al., 2023).

In opposition to implying relevant biases in neural networks, recent studies suggest a data-driven approach to learning the dynamics of one type of system generalized across different environments (Yin et al., 2021; Kirchmeyer et al., 2022). These focus on dynamical systems that are governed by the form of the following differential equation; $\frac{d\boldsymbol{x}_e(t)}{dt} = f_e(\boldsymbol{x}_e(t))$, where $\boldsymbol{x}(t)$ is a time-dependent state in the phase space $\mathcal{X}$, and $f_e : \mathcal{X} \to T\mathcal{X}$ maps $\boldsymbol{x}_e(t)$ to its time-derivative state $\frac{d\boldsymbol{x}_e(t)}{dt}$ which lies in the tangent space $T\mathcal{X}$. Here, the subscript $e$ denotes an environment for a given dynamics (e.g. different $\alpha, \beta, \gamma, \delta$ parameter configuration of Lotka-Volterra system; $du/dt = \alpha u - \beta uv, dv/dt = \delta uv - \gamma v$).

### 2.3 META LEARNING FOR GENERALIZING DYNAMICAL SYSTEM

Meta learning, also referred to as "learning to learn," focuses on training a model that exhibits strong generalization across diverse data distributions, enabling it to effectively adapt to new tasks even with a small amount of previously unseen samples. As a result, meta learning can serve as a promising data-driven approach for generalizing the dynamics of physical systems. Among various approaches of meta learning, optimization-based methods are particularly versatile and compatible with differentiable models (Hospedales et al., 2020). For example, CoDA (Kirchmeyer et al., 2022) is a representative optimization-based meta learning method for multi-environment formulation of the generalization problem for dynamical systems. Particularly, previous studies (Lee et al., 2021; Li et al., 2023) have demonstrated the ability of a broadly applicable optimization-based meta-learning algorithm, Model-Agnostic Meta-Learning (MAML) (Finn et al., 2017), in uncovering the physical laws of a given system across different environments. There also exists non-MAML based methods for generalizing dynamics. Sequential Neural Processes (Singh et al., 2019) aims to build a meta-transfer learning framework for stochastic processes, DyAd (Wang et al., 2022) is a model-based meta learning method that learns the shared dynamics of a given domain, and meta-SLVM (Jiang et al., 2022) utilizes Bayesian meta learning for learning to adapt latent dynamic functions.

The approaches mentioned above are all limited to scenarios where the systems share the same functional form of Hamiltonians, such as a set of pendulum systems with varying physical parameters (e.g. mass and length). Expanding upon these limitations, we aim to investigate whether a shared neural representation can be discovered among diverse functional forms of Hamiltonians. In this study, we explore the portrayal of a symplectic manifold governed by Hamilton's equation within neural networks by incorporating MAML with GNNs. Further details regarding the MAML algorithm can be found in Section 3.2 and 3.3.

## 3 METHODS

### 3.1 PREPARING THE DATASET FOR META LEARNING

We constructed a task distribution comprising six distinct types of physical systems: mass-spring, pendulum, Hénon-Heiles, magnetic-mirror, two-body, and three-body. Given that these systems feature separable Hamiltonians, which inherently limit the diversity in the kinetic energy terms, we intentionally selected systems with varying complexities in their potential energy terms. Starting from the simplest mass-spring system, we extend the type of system to more difficult systems such as Hénon-Heiles or magnetic-mirror system. The details for each system can be found in the Supplementary Materials (SM) A.1.

This variety of systems encompasses a range of complexities, featuring simple physical systems like the mass-spring and the pendulum, as well as more intricate systems exhibiting chaotic behavior such as the Hénon-Heiles and the magnetic-mirror. Additionally, the task distribution extends to systems involving more than two particles, specifically the two-body, and the three-body, thereby providing a comprehensive set of challenges. For all of our systems, we generated a dataset with $N = 10,000$ trajectories confined to the two-dimensional space, where the canonical coordinates $\boldsymbol{x} = (\boldsymbol{q}, \boldsymbol{p})$ as the input, and their corresponding time derivatives $\dot{\boldsymbol{x}} = (\dot{\boldsymbol{q}}, \dot{\boldsymbol{p}})$ as the output of a neural network. We provide the detailed procedure for generating the dataset in the SM A.2.

In the language of meta learning, several different types of systems will be used for meta training, while one of the remaining unselected systems will be used for meta testing (i.e. evaluation). We will denote each trajectory as $\mathcal{T}_{\mathbb{S},i}$, which is sampled from the task distribution $p(\mathcal{T}_{\mathbb{S}})$ (where $S \in$ {mass-spring, pendulum, Hénon-Heiles, magnetic-mirror, two-body, three-body}, and $i \in [\![1..N]\!]$). Then tasks from $\mathcal{T}_{\mathbb{S}^{\complement}}$ will compose the task distribution for meta training, and tasks from $\mathcal{T}_{\mathbb{S}}$ will form the data distribution for meta testing. For example, we will use the data distribution $\mathcal{T}_{\mathbb{S}^{\complement}}$; $\mathbb{S}^{\complement} = $ {mass-spring, pendulum, Hénon-Heiles} for meta training, $\mathcal{T}_{\mathbb{S}}$; $S = $ {magnetic-mirror} for evaluating the magnetic-mirror system. The exact dataset configuration for meta training scenario is described in SM A.3.

## 3.2 META TRAINING THE NEURAL NETWORK

We use a simple graph convolutional network (GCN) (Kipf & Welling, 2016) to parametrize the given physical system in which we implement it using PyTorch (Paszke et al., 2019). Using GNNs to describe a system of several particles mediates the model to handle various degree-of-freedom inputs such that the model can be fed with data from a diverse type of system, unlike from the previous works listed in Section 2.2, and 2.3 that dealt with fixed system scenarios. In line with the previous methods (Sanchez-Gonzalez et al., 2019; Bishnoi et al., 2023) that represented the physical systems as graphs, we represent the state variable $(\boldsymbol{q}, \boldsymbol{p})$ as the node feature, while currently, edge features are not utilized. Our model is composed of three GCN layers to extract features of the input states, preceded by three fully connected linear layers for the regression of Hamiltonian value. We choose the *mish* function as the non-linear activation function. Prior works (Greydanus et al., 2019; Sanchez-Gonzalez et al., 2019; Lee et al., 2021) have utilized *tanh* or *softplus* activations, hypothesizing that the *relu* activation may hinder parameter optimization due to its piecewise linear nature, as the HNN loss defined in Equation 2 involves calculating derivatives of the output with respect to the inputs. After conducting several trials, we observed that the *mish* activation produced more stable and improved results compared to *tanh* or *softplus* activations in our case. More details regarding the network architecture are further described in the SM B.1.

Upon the framework that we described so far, we represent our model as $f_\theta$; parameterized by neural network parameters $\theta$. To account for the varying input ($\boldsymbol{x}^i$) and output ($\dot{\boldsymbol{x}}^i$) scales across different systems, we employ the log-cosh function as our loss function. This choice enhances the robustness of our model to data with diverse system dynamics, as it mitigates the impact of differing data scales.

$$\mathcal{L}_{\mathcal{T}_i}(f_\theta) = \sum_{(\boldsymbol{x}^i, \dot{\boldsymbol{x}}^i) \sim \mathcal{T}_i} \log \cosh\left(\dot{\boldsymbol{x}}^i - J\nabla_{\boldsymbol{x}} f_\theta(\boldsymbol{x}^i)\right) \tag{3}$$

In the inner loop of our variation of MAML, the model adapts to a new task $\mathcal{T}_{\mathbb{S}^{\complement},i}$ by updating the parameters with one gradient step using $K = 50$ phase space points for each task.

$$\theta'_i = \theta - \alpha\nabla_\theta \mathcal{L}_{\mathcal{T}_{\mathbb{S}^{\complement},i}}(f_\theta) \tag{4}$$

In the outer loop using the Adam optimizer (Kingma & Ba, 2014), meta optimization across systems is done by updating the parameters as below.

$$\theta \leftarrow \theta - \beta\nabla_\theta \sum_i \mathcal{L}_{\mathcal{T}_{\mathbb{S}^{\complement},i}}(f_{\theta'_i}) \tag{5}$$

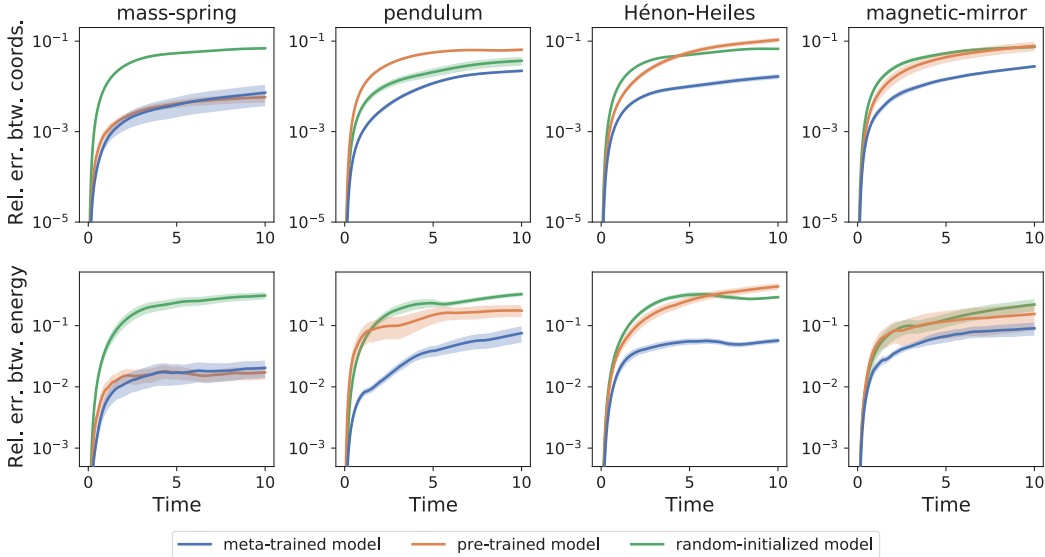

Figure 1: The relative error throughout the time rollout for the meta-trained, pre-trained, and random-initialized model at adaptation step 50 for the predicted coordinates (top) and energy (bottom). The solid line and the shaded area each represent the average and the standard error of 10 runs.

## 3.3 EVALUATING THE META-TRAINED MODEL

We evaluate the performance by making a few-step gradient descent on the unseen type of system during the meta training phase with $K = 50$ data points. The model adaptation to the unseen system is done by updating the parameters as follows.

$$\theta \leftarrow \theta - \alpha \nabla_\theta \mathcal{L}_{\mathcal{T}_{\mathbb{S},i}}(f_\theta) \tag{6}$$

Using the adapted model at each step, we integrate our model output using the LSODA integrator (Petzold, 1983) to achieve the system dynamics. Then, the performance was achieved by calculating the relative error between the predicted and ground truth trajectory points. We perform our experiments on NVIDIA RTX A6000 by performing 10 independent runs as default to ensure the stability of our results and report the average and standard error values. We again further describe the related details for the training and adaptation process in the SM B.2.

## 4 RESULTS

## 4.1 QUANTITATIVE ANALYSIS OF THE MODEL PERFORMANCE

For each system, we evaluated the adaptation performance on unseen systems for the meta-trained model and compared it with the pre-trained model that doesn't utilize meta learning, and the scratch model which was adapted from random-initialized weights. We conducted up to 50 adaptation steps and computed the relative error between the predicted and the ground truth coordinates and energies at each time rollout and adaptation step. The measured relative error is defined as $\text{Err}(t) = \|\hat{z}(t) - z(t)\|_2 / (\|\hat{z}(t)\|_2 + \|z(t)\|_2)$, as described in (Finzi et al., 2020). This error metric enables us to quantify the error independent of the data scale and tends towards unity as the predictions become orthogonal to the ground truth or $\|\hat{z}\| \gg \|z\|$. Compared to metrics like mean squared error, the above notion of bounded relative error provides a more accurate representation of the prediction performance. Finally, we calculated the geometric moving average of the relative error to account for the multiplicative accumulation of the error from the numerical integration involved.

In Figure 1, the relative error in the predicted trajectories serves as an indicator of the overall accuracy of the adapted models, while the energy metric demonstrates that the physics of the system remains well-behaved during the prediction. Considering both predicted coordinates and energy, our results demonstrate that the meta-trained model robustly outperforms the other two baselines.

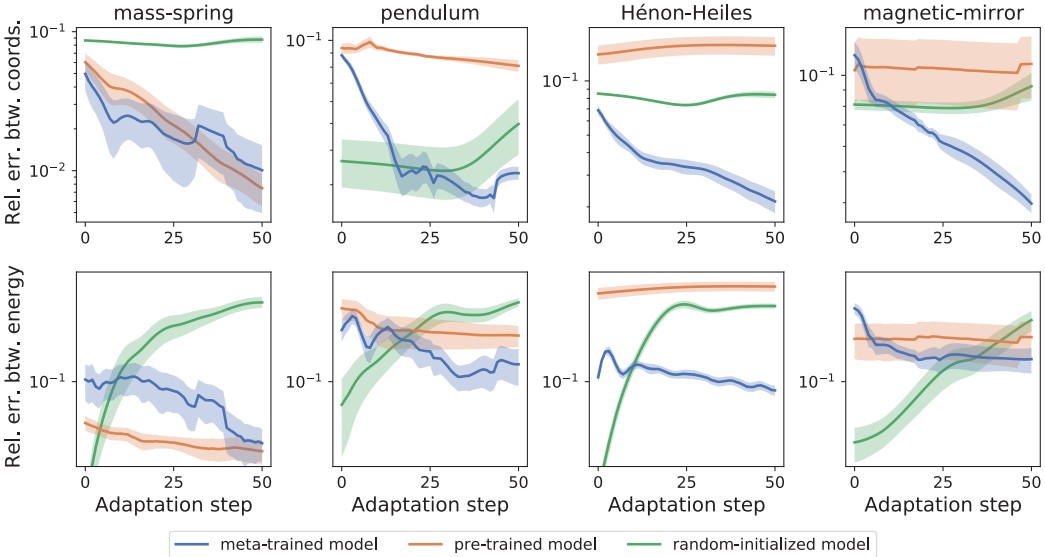

Figure 2: The relative error across the adaptation step for the meta-trained, pre-trained, and random-initialized model at adaptation step 50 for the predicted coordinates (top) and energy (bottom). The solid line and the shaded area each represent the average and the standard error of 10 runs.

We further examine the capability of fast adaptation of each model throughout the adaptation steps. Figure 2 illustrates the decreasing trend of the relative error as the number of adaptation steps increases from 0 to 50. Note that the meta-trained model requires significantly fewer adaptation steps compared to the randomly initialized model, which corresponds to the vanilla HNN model. Also, the relative error between the predicted coordinates shows that the fast adaptation shows a significant difference as the system becomes more complex. Within a gradient step of less than 100, the meta-trained model demonstrates faster and superior adaptation compared to both the pre-trained model without meta learning and the HNN baseline.

## 4.2 QUALITATIVE ANALYSIS OF THE PREDICTED DYNAMICS

In Section 4.1, we observed that the meta-trained model consistently achieved a lower relative error between the predicted and ground truth dynamics across both time rollouts and adaptation steps. However, a small error in the predicted trajectory does not always guarantee accurate system dynamics prediction. To provide more conclusive results, we examined the exact predicted dynamics of the three adapted models after 50 steps. We conducted this analysis for the mass-spring, pendulum, Hénon-Heiles, and magnetic-mirror systems, and the corresponding results are shown in Figure 3.

For the mass-spring system, both the meta-trained model and pre-trained model accurately predicted the coordinates $(x, p_x)$ within 50 adaptation steps, while the random-initialized model required more steps to adapt. Here, it is worth noting the behavior of the prediction of the surplus coordinates $(y, p_y)$. In the real world, we cannot readily distinguish what coordinates remain physically meaningful among different phase variables. Thus, it would be beneficial if our trained neural network could also distinguish which one is necessary for describing the system. In the experiments, we confined the dynamics of the system to a two-dimensional space, thus $y, p_y$ should remain constant for the mass-spring system. Comparing the predicted trajectories of the redundant coordinates between the meta-trained model and the pre-trained model, we observed that the meta-trained model had a much smaller scale in the prediction of unnecessary coordinates $(y, p_y)$.

Moving on to the pendulum system, the meta-trained model also provides better predictions than the other two baselines. While the pre-trained model performed better than the random-initialized model, the dynamics of $\theta$ and $p_\theta$ were still out of phase compared to the meta-trained model. We also observed a similar behavior in the prediction of the surplus coordinates $(r, p_r)$, as observed in the mass-spring system.

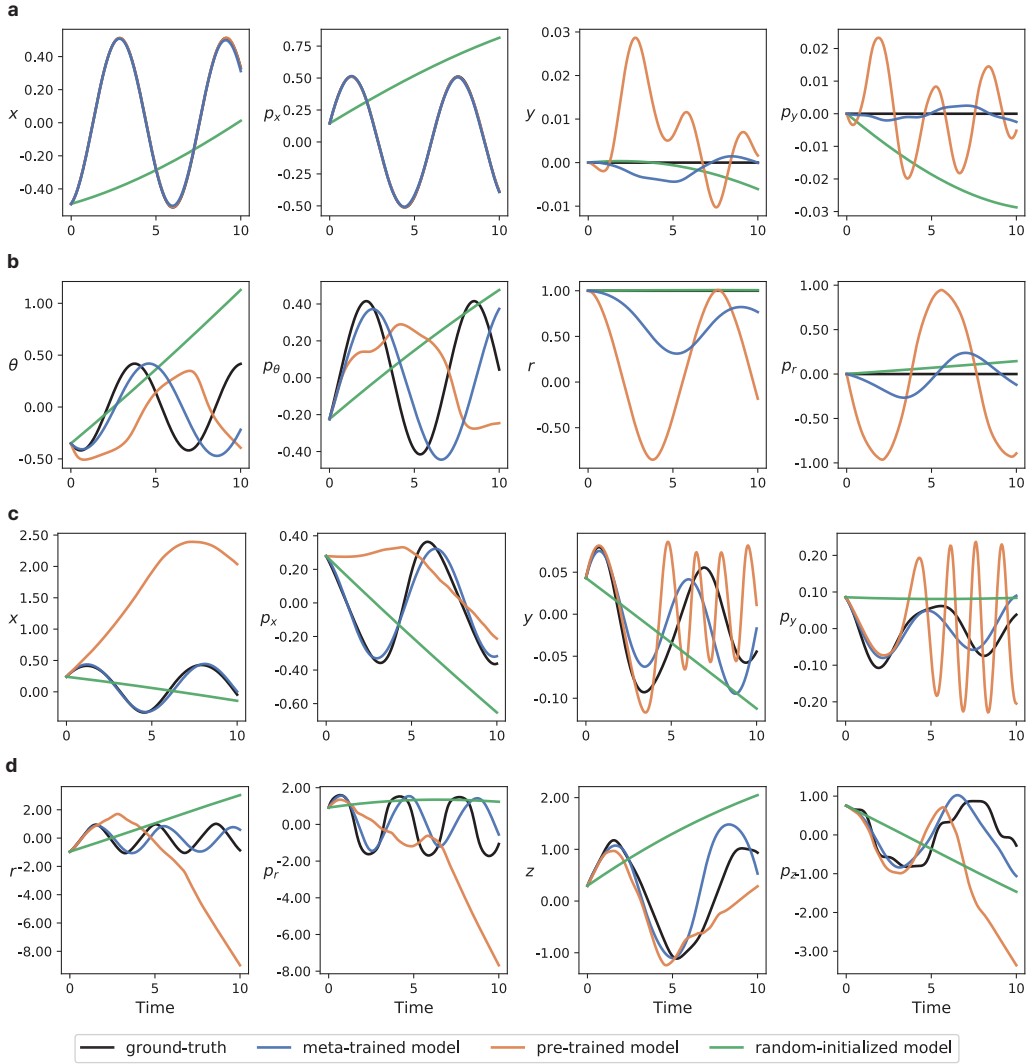

Figure 3: System dynamics through time rollout 0 to 10, generated from the meta-trained, pre-trained, and random-initialized model each after 50 adaptation steps. Each row (a) to (d) corresponds to the predictions with mass-spring, pendulum, Hénon-Heiles, and the magnetic mirror system respectively.

For the more complex systems exhibiting chaotic behavior, such as the Hénon-Heiles and magnetic-mirror system, all four coordinates are necessary to describe the dynamics. In these scenarios, the meta-trained model significantly outperformed the pre-trained model and random-initialized model. Although the meta-trained model did not perfectly match the trajectory, it captured the overall behavior of each coordinate dynamics more reasonably compared to the other baselines.

## 4.3 ANALYSIS ON THE LEARNED REPRESENTATION

Our objective in this section is to understand why the meta-trained model outperforms the other baselines, particularly the pre-trained model. The existing difference between the meta-trained model and the pre-trained model lies in the training strategy during the training process, while factors such as the total iterations, along with other training conditions remained the same. To investigate the impact of the training algorithm, we conducted the centered kernel alignment (CKA) (Kornblith et al., 2019) analysis on the three models. CKA offers a means to compare the representations of two layers within a neural network, providing a similarity score ranging from 0 to 1. Furthermore, it is widely recognized that early layers tend to learn general representations, while later layers tend to specialize in specific features(Yosinski et al., 2014; Raghu et al., 2019; 2020; Alabdulmohsin et al., 2021).

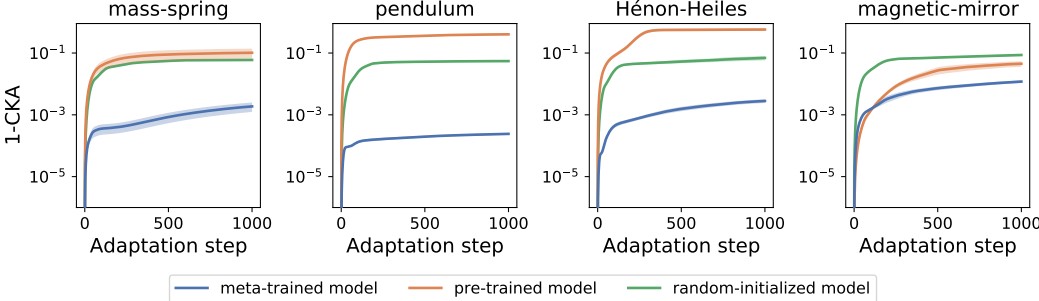

Figure 4: The measured CKAs in the last layer between the model before adapting (i.e. right after finishing meta-training or pre-training), and during the adaptation up to 1000 steps. The CKA values are transformed to 1-CKA to make a clearer visualization. The solid line and the shaded area each represent the average and the standard error of 10 runs.

Upon this observation, we examine the CKA between the model's last layer before adaptation and its state at each adaptation step, ranging from 1 to 1000. Figure 4 shows these 1000 individual 1-CKA measurements, each comparing the model before adaptation to its state at a specific step during the adaptation process. In all four cases, the meta-trained model consistently exhibited substantially lower 1-CKA values compared to the other models. This finding suggests that the representations learned during the adaptation process closely resemble those of the meta-trained model when it serves as the starting point. Drawing from insights gained in the prior works utilized meta learning (Lee et al., 2021; Li et al., 2023), and the manifold hypothesis (Fefferman et al., 2016), we interpret the meta-trained model gives the representation of the fundamental structure of underlying Hamiltonian dynamics. Thus making the meta-trained model acquire a generalized understanding of the diverse domain of distinct physical law. During the evaluation, the model adeptly adapts to unknown dynamics by leveraging the generalized physics learned during the meta training phase. Consequently, this observation provides further support for our initial assumption that meta learning can capture general representations across a broad domain of Hamiltonian manifolds.

### 4.4 Ablation analysis and comparison with existing methods

First, we present the ablation studies to scrutinize the impact of both the choice of integrator and the number of data points $K$ in the adaptation task. We experimented with symplectic integrators ranging from 1st to 4th order to ensure energy conservation compared to the LSODA integrator employed in our primary experiments. And, since we are considering scenarios where abundant data is available for known systems but sparse data for the target unknown system, we varied the test data size as $K \in (5, 10, 20, 50)$ to investigate the data efficiency of the meta-trained model relative to other baselines. The ablation results for varying integrator and $K$ each detailed in SM C.1 and C.2, reveal several key findings. 1) LSODA integrator performs adequately for dynamics that do not involve large time scales, making it a reliable choice for most scenarios considered in this study. 2) Meta-trained model demonstrates the data efficiency when adapting to unseen tasks. This suggests that the meta-trained model is not only data-efficient but also robust, capable of adapting well when data for the target system is sparse.

Second, we demonstrate the generalization ability of the meta-trained model by not just testing the adaptation to a single-held-out system scenario as in the previous sections but also checking the adaptation performance on various types of unseen systems (i.e. performing the adaptation task to multi-held-out-system scenario). This task involved four models, each of which was initially trained for adaptation to one of the following systems; mass-spring, pendulum, Hénon-Heiles, and magnetic-mirror. We conducted an additional adaptation task using namely the 2D harmonic oscillator system and the Kepler system. We evaluated the performance of these models in terms of relative error for both predicted coordinates and energy, with detailed results presented in SM C.3. The results indicate that the meta-trained model is capable of effectively adapting to more than just a single type of unseen system, further suggesting the potential of a general model for real-world applications. This underscores the robustness and versatility of the generalized representation learned by the meta-trained model.

Third, we compared the meta-trained model to the existing baselines in domain generalization for learning the physical dynamics. Although there is none that aims to generalize across the type of dynamical systems as we have already discussed in Section 2.3, it is important to recognize that the methods from previous works are not appropriate in our broader scope of generalization. We considered CoDA (Kirchmeyer et al., 2022), which showed the best performance across prior methods for generalizing dynamics across multi-environment within a fixed system setting, and DyAd (Wang et al., 2022), which is a representative example of a non-optimization-based (model-based) meta-learning method as our baselines. The results in SM C.4, and C.5 show that the meta-trained model outperforms the CoDA and DyAd baselines for the generalization problem in cross domain settings. Moreover, note that in some cases, the existing baselines struggled to adapt.

## 4.5 LIMITATIONS OF OUR APPROACH

In our experiments, we treated all systems as conservative. However, data from unknown physics may not satisfy these conditions, raising concerns about its applicability to real-world scenarios. Correspondingly, we performed the adaptation task to a damped mass-spring system with a damping term $-c\dot{x}$. From the results in SM C.6, we can see that the non-conservative feature disturbs the adaptation. This is due to the damping term $-c\dot{x}$, Equation 1 is violated, which hinders the validity of Equation 2 during adaptation. Thus to make use of our approach, both the data used in meta training and adaptation should not differ in their intrinsic nature.

Furthermore, we extend our analysis to two-body and three-body systems, going beyond the scenarios on single-particle tasks, thereby opening the door to real-world applications. This extension allows us to explore the dynamics of two-body systems as a direct continuation, and three-body systems as a more challenging problem due to their chaotic behavior. However, we acknowledge that further refinement is required for the meta learning configuration for non-single-particle tasks. Here, we present the preliminary results for the two-body and three-body systems in the SM C.7.

## 5 DISCUSSION

While previous studies have focused on generalizing system dynamics across different environments under consistent physical laws, our work takes a step further by aiming for generalization across diverse system domains. Specifically, we tackle the challenge of identifying the shared representation of physical systems across various functional forms of Hamiltonian. In this perspective, our work presents unique contributions compared to existing literature. Here, we leverage the power of meta learning algorithm to uncover the representation of the Hamiltonian manifold within neural networks. In our comparison between the meta-trained model and other baseline models, we provide compelling evidence that the neural network learns the Hamiltonian of an unseen system by identifying the general representation for Hamiltonian dynamics itself, rather than directly constructing the functional form of the Hamiltonian of the given system. This finding highlights the ability of meta learning to capture and exploit the underlying structure of Hamiltonian mechanics. To support our claim, we conduct a CKA analysis on the trained networks. This analysis provides valuable insights into why the meta-trained model outperforms the other baselines across different physics. By examining the similarity between representations before and during the adaptation process, we gain an understanding of how the meta-trained model effectively learns and adapts to diverse physical systems. Overall, our work contributes to the field by demonstrating the efficacy of meta learning in discovering shared representations of physical systems across distinct functional forms of Hamiltonian.

In nature, each physical system operates according to its unique governing law, which is rooted in fundamental principles. In the scope of this work, systems with different functional forms of Hamiltonian are expressed with one Hamilton's equation. Building upon this observation, we propose an approach that harnesses the power of meta learning algorithms to embed these fundamental physical principles into neural networks. Our method allows us to capture and represent these general physical principles within the neural network architecture, thereby providing a means to interpret neural networks in a physical context. A key advantage of our approach is that we imply minimal predefined physical priors or inductive biases. Instead, it leverages the data-driven capabilities of deep learning to uncover the underlying physics. This marks a significant step forward in developing a more comprehensive and general model for discovering physics through the lens of neural networks.

## ACKNOWLEDGMENTS

This research was supported by the Basic Science Research Program through the National Research Foundation of Korea NRF-2022R1A2B5B02001752.

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

# A DATASET SPECIFICATION

## A.1 TYPES OF SYSTEMS USED

Below are the descriptions of the systems that are used for the dataset.

**Mass-Spring** One of the most simplest physical system is one particle, frictionless mass-spring system, where $m, k$ are the mass of the particle, and spring constant respectively.

$$\mathcal{H} = \frac{p_x^2}{2m} + \frac{kx^2}{2}$$

**Pendulum** The Hamiltonian of a pendulum system is slightly more complex than the mass-spring case. Here, $m, g, l$ denotes the mass of the particle, gravitational acceleration, and length of the pendulum respectively.

$$\mathcal{H} = \frac{p_\theta^2}{2ml^2} + mgl(1 - \cos\theta)$$

**Hénon-Heiles** Up to here, the given dynamical system was rather simple. Hénon-Heiles gives the chaotic dynamics of a star around a galactic center with its motion constrained on a 2D plane (Hénon & Heiles, 1964; Hénon, 1983). In the below Hamiltonian of the Hénon-Heiles system, $\lambda$ is conventionally taken as unity.

$$\mathcal{H} = \frac{1}{2}(p_x^2 + p_y^2) + \frac{1}{2}(x^2 + y^2) + \lambda\left(x^2 y - \frac{y^3}{3}\right)$$

**Magnetic-Mirror** Here we introduce a system that has the most complicated form of Hamiltonian in our experiments. From the works of (Efthymiopoulos et al., 2015), the Hamiltonian of the magnetic bottle type system is given as follows.

$$\mathcal{H} = \frac{1}{2}(\dot{\rho}^2 + \dot{z}^2) + \frac{1}{2}\rho^2 + \frac{1}{2}\rho^2 z^2 - \frac{1}{8}\rho^4 + \frac{1}{8}\rho^2 z^4 - \frac{1}{16}\rho^4 z^2 + \frac{1}{128}\rho^6$$

**Two-Body** From now on, we expand our system with more than two particles. In the two-body system case, we consider the gravitational interaction between two particles. Then the Hamiltonian of the two-body system can be written as follows. Note that $G$ is the gravitational constant, and $m_1, m_2$ are the masses for each of the two bodies.

$$\mathcal{H} = \frac{\boldsymbol{p}_1^2}{2m_1} + \frac{\boldsymbol{p}_2^2}{2m_2} - \frac{Gm_1 m_2}{|\boldsymbol{r}_1 - \boldsymbol{r}_2|}$$

**Three-Body** Adding a particle to the two-body system gives the three-body system. Although the Hamiltonian of the three-body system is an incidental extension of the two-body case, its dynamics cannot be described by a closed-form expression, thus exhibits chaotic behaviour.

$$\mathcal{H} = \frac{\boldsymbol{p}_1^2}{2m_1} + \frac{\boldsymbol{p}_2^2}{2m_2} + \frac{\boldsymbol{p}_3^2}{2m_3} - \frac{Gm_1 m_2}{|\boldsymbol{r}_1 - \boldsymbol{r}_2|} - \frac{Gm_2 m_3}{|\boldsymbol{r}_2 - \boldsymbol{r}_3|} - \frac{Gm_3 m_1}{|\boldsymbol{r}_3 - \boldsymbol{r}_1|}$$

## A.2 DATASET GENERATION

Using the Hamiltonian described in Section 3.1, we obtained the state trajectories $(\boldsymbol{q}, \boldsymbol{p})$ by employing the LSODA integrator implemented in SciPy (Virtanen et al., 2020). The trajectories were integrated over the interval $[0, 10]$ with 200 steps. Subsequently, we calculated the corresponding time derivatives $(\dot{\boldsymbol{q}}, \dot{\boldsymbol{p}})$ using JAX (Bradbury et al., 2018) and Equation 1. For simplicity, all the constants from the Hamiltonian (i.e., $m_i, k, l_i, g, G$) were set to 1. The initial conditions employed for each system are described below.

**Mass-Spring** The initial state $(x_0, p_{x0})$ is randomly sampled from a uniform distribution over the interval $[-1, 1]^2$. The redundant coordinate $(y_0, p_{y0})$ is set to $(0, 0)$ as a fixed value.

**Pendulum** The initial state $(\theta_0, p_{\theta 0})$ is randomly sampled from a uniform distribution over the interval $\left[-\frac{\pi}{2}, \frac{\pi}{2}\right] \times [-1, 1]$. The redundant coordinate $(r_0, p_{r0})$ is set to $(1, 0)$ as a fixed value.

**Hénon-Heiles** The initial state $(x_0, y_0, p_{x0}, p_{y0})$ is randomly sampled from a uniform distribution over the interval $[-1, 1]^4$.

**Magnetic-Mirror** The initial state $(\rho_0, z_0, p_{\rho_0}, z_0)$ is randomly sampled from a uniform distribution over the interval $[-1, 1]^4$.

**Two-Body** The initial coordinate for the first body $(x_{1_0}, y_{1_0})$ is randomly sampled from a uniform distribution over the interval $[0.5, 1.5]^2$, and $(p_{x_{1_0}}, p_{y_{1_0}})$ is calculated to obtain a nearly-circular orbit. Then, the initial state for the second body $(x_{2_0}, y_{2_0}, p_{x_{2_0}}, p_{y_{2_0}})$ is set to $(-x_{1_0}, -y_{1_0}, -p_{x_{1_0}}, -p_{y_{1_0}})$ Then we slightly perturbed $p_x, p_y$ by adding a Gaussian noise (multiplied by a constant of 0.1) to the velocity of both two bodies. Here, the velocities are equivalent to the canonical momentum $p_x$ and $p_y$, as we assume the masses $m_1$ and $m_2$ to be equal to 1.

**Three-Body** The initial state for the three-body system is obtained in a similar way to that of the two-body, except the initial coordinate for the first body $(x_{1_0}, y_{1_0})$ is randomly sampled from a uniform distribution over the interval $[0.8, 1.2]^2$. Again, $(p_{x_{1_0}}, p_{y_{1_0}})$ is set to obtain a nearly-circular orbit. The initial state for the second and third body is obtained by rotating the first and second body each by an angle of $\frac{2\pi}{3}$. Here, the Gaussian noise term that is added to each of the bodies is multiplied by a constant of 0.05.

Following the above description, we generated 10000 trajectories for each system.

### A.3 DATASET CONFIGURATION FOR META TRAINING

The composition of the system types used in our meta training scenario is listed in Table 1.

Table 1: Dataset configuration for meta learning Hamiltonian systems

| system to test | systems for meta training |
|---|---|
| mass-spring | pendulum, Hénon-Heiles, magnetic-mirror |
| pendulum | mass-spring, Hénon-Heiles, magnetic-mirror |
| Hénon-Heiles | mass-spring, pendulum, magnetic-mirror |
| magnetic-mirror | mass-spring, pendulum, Hénon-Heiles |
| two-body | mass-spring, pendulum, Hénon-Heiles, magnetic-mirror |
| three-body | mass-spring, pendulum, Hénon-Heiles, magnetic-mirror |

## B NEURAL NETWORK IMPLEMENTATION

### B.1 NETWORK ARCHITECTURE

In our experiments, we constructed our model as follows: GCNConv(4, 200) - Mish - GCNConv(200, 200) - Mish - GCNConv(200, 4) - Mish - GlobalMeanPool - Linear(4, 200) - Mish - Linear(200, 200) - Mish - Linear(200, 1) - Mish, where GCNConv and Linear correspond to the graph convolutional layer and the fully connected layer implemented in PyTorch respectively. Thus, the model outputs a single-dimensional scalar value. After the forward pass, the derivative of the output with respect to the input is computed to obtain the time-derivative of the input state, utilizing Equation 1. Note that we did not heavily tune the hyperparameters regarding the network architecture, because we focus on the difference between the meta-trained, pre-trained, and random-initialized model, not obtaining state-of-the-art results.

### B.2 TRAINING PROCESS

**Mass-Spring** For both meta-training and pre-training, we used a learning rate of $\alpha = 0.001$ for the gradient step on the inner loop, and the Adam optimizer on the outer loop with a learning rate of $\beta = 0.0005$. The inner gradient update was 1 step, with a total of 5000 iterations on the outer loop. The number of task batches is set to 10, and the number of phase points used for each task was 50 (i.e. among the 200 points in each trajectory, 50 points were randomly sampled for meta training). For evaluation, we used the Adam optimizer with a learning rate of 0.0001.

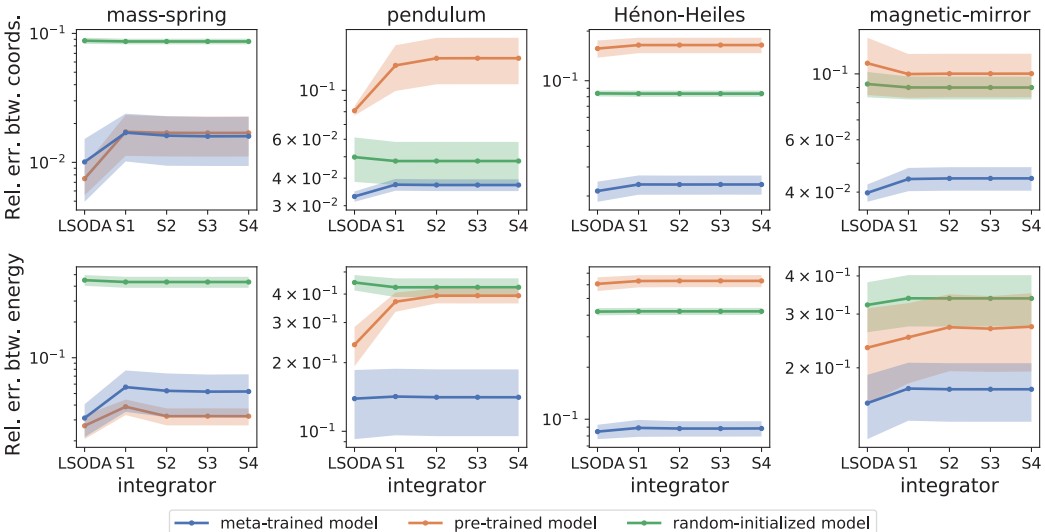

Figure S.1: The relative error for the meta-trained, pre-trained, and random-initialized model with respect to the predicted coordinates (top) and energy (bottom) using different integrators. The solid line and the shaded area each represent the average and the standard error of 10 runs.

**Pendulum** All conditions were set to be the same as those of the mass-spring system.

**Hénon-Heiles** All conditions were set to be the same as those of the mass-spring system, except for the iterations on the outer loop which was set to 10000.

**Magnetic-Mirror** All conditions were set to be the same as those of the mass-spring system, except for the iterations on the outer loop which was set to 30000.

**Two-Body** For both meta-training and pre-training, we used a learning rate of $\alpha = 0.001$ for the gradient step on the inner loop, and the Adam optimizer on the outer loop with a learning rate of $\beta = 0.0005$. We set 10000 iterations on the outer loop. The other conditions were set to be the same.

**Three-Body** All conditions were set to be the same as those of the two-body system, except for the $\beta$ which is set to $0.0005$.

## C ADDITIONAL EXPERIMENTS

### C.1 ABLATION ON INTEGRATORS

Figure S.1 shows the evaluation performance of meta-trained, pre-trained, and randomly-initialized models across scenarios using different integrators; LSODA, and the 1st~4th order symplectic integrator. The LSODA integrator consistently yields the best results in most cases. While symplectic integrators are known for their ability to maintain energy conservation, their advantages become more apparent in long-term dynamics. However, the time range considered in this study is not sufficient to fully capture the benefits of symplectic integrators. Nonetheless, their impact could be significant when extended to longer time scales. So in our experiments, we use LSODA as our primary integrator.

### C.2 ABLATION ON $K$

Figure S.2 presents the evaluation performance of meta-trained, pre-trained, and randomly-initialized models under varying $K$ in the adaptation task. The ablation results indicate that the meta-trained model consistently performs well at low $K$ values across adaptation tasks. Furthermore, its performance markedly improves as the data size $K$ increases, which reflects the efficient adaptability of the meta-trained model.

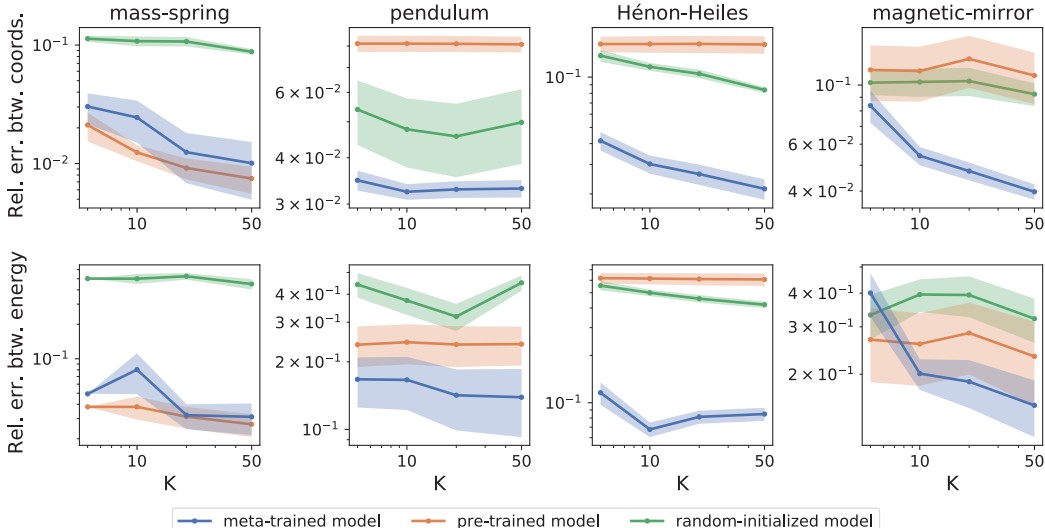

Figure S.2: The relative error for the meta-trained, pre-trained, and random-initialized model with respect to the predicted coordinates (top) and energy (bottom) with various number of data points $K$ used. The solid line and the shaded area each represent the average and the standard error of 10 runs.

### C.3 GENERALIZATION FOR MULTI-HELD-OUT SYSTEM

Here, we present the adaptation performance of the meta-trained models in Section 4 to the multi-held-out system scenario for the extent of the single-held-out scenario demonstrated in Section 4. For example, we performed the adaptation task of the meta-trained model trained with a pendulum, Hénon-Heiles, and the magnetic-mirror system not only for the mass-spring system, but also with the newly introduced 2D harmonic oscillator, and the Kepler system as well.

The relative errors in terms of both predicted coordinates and energy for this test are presented in Figure S.3, and S.4. The results demonstrate that the meta-trained models exhibit proficient adaptation both to the newly introduced 2D harmonic soscillator system and the Kepler system.

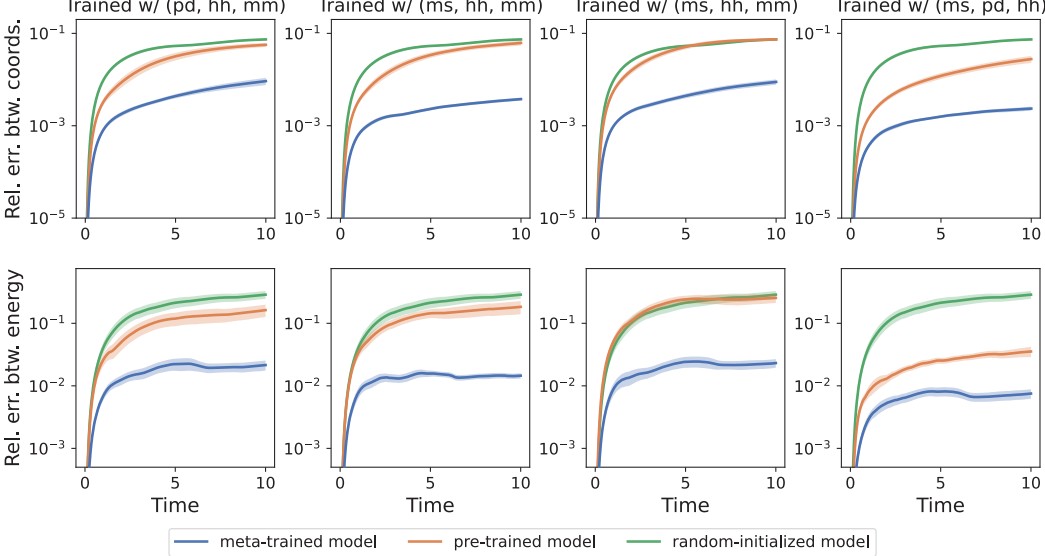

Figure S.3: The relative error throughout the time rollout of the 2D harmonic oscillator system for the meta-trained, pre-trained, and random-initialized model at adaptation step 50 with respect to the predicted coordinates (top) and energy (bottom). The solid line and the shaded area each represent the average and the standard error of 10 runs.

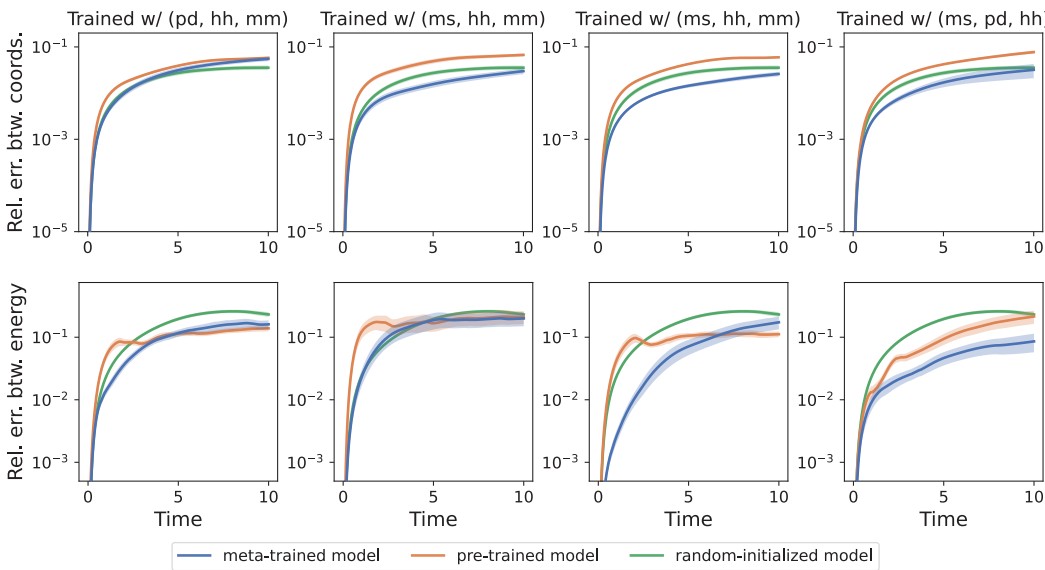

Figure S.4: The relative error throughout the time rollout of the Kepler system for the meta-trained, pre-trained, and random-initialized model at adaptation step 50 with respect to the predicted coordinates (top) and energy (bottom). The solid line and the shaded area each represent the average and the standard error of 10 runs.

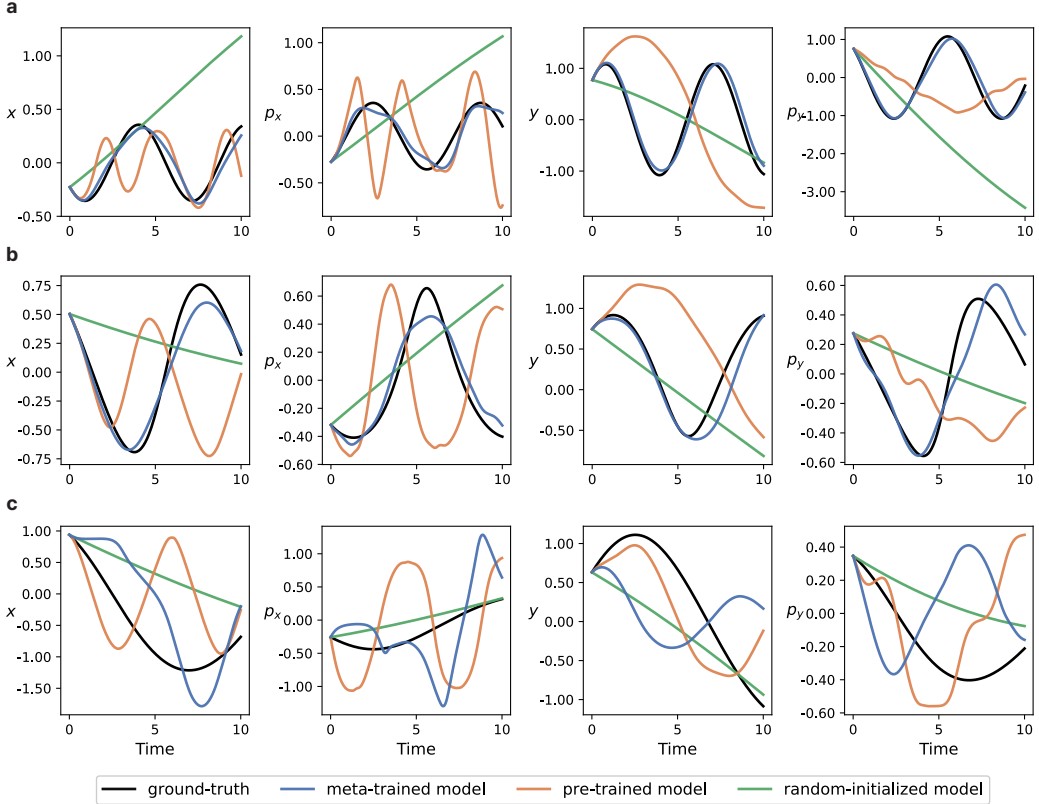

Figure S.5: The example of predicted dynamics for the 2D harmonic oscillator system (a), and each good (b) and bad (c) examples of Kepler system.

We also show the example of the predicted dynamics for both the 2D harmonic oscillator and the Kepler system in Figure S.5. There exists a situation for the Kepler system where the meta-trained model does not perform well, which implies insufficient adaptation steps for this case.

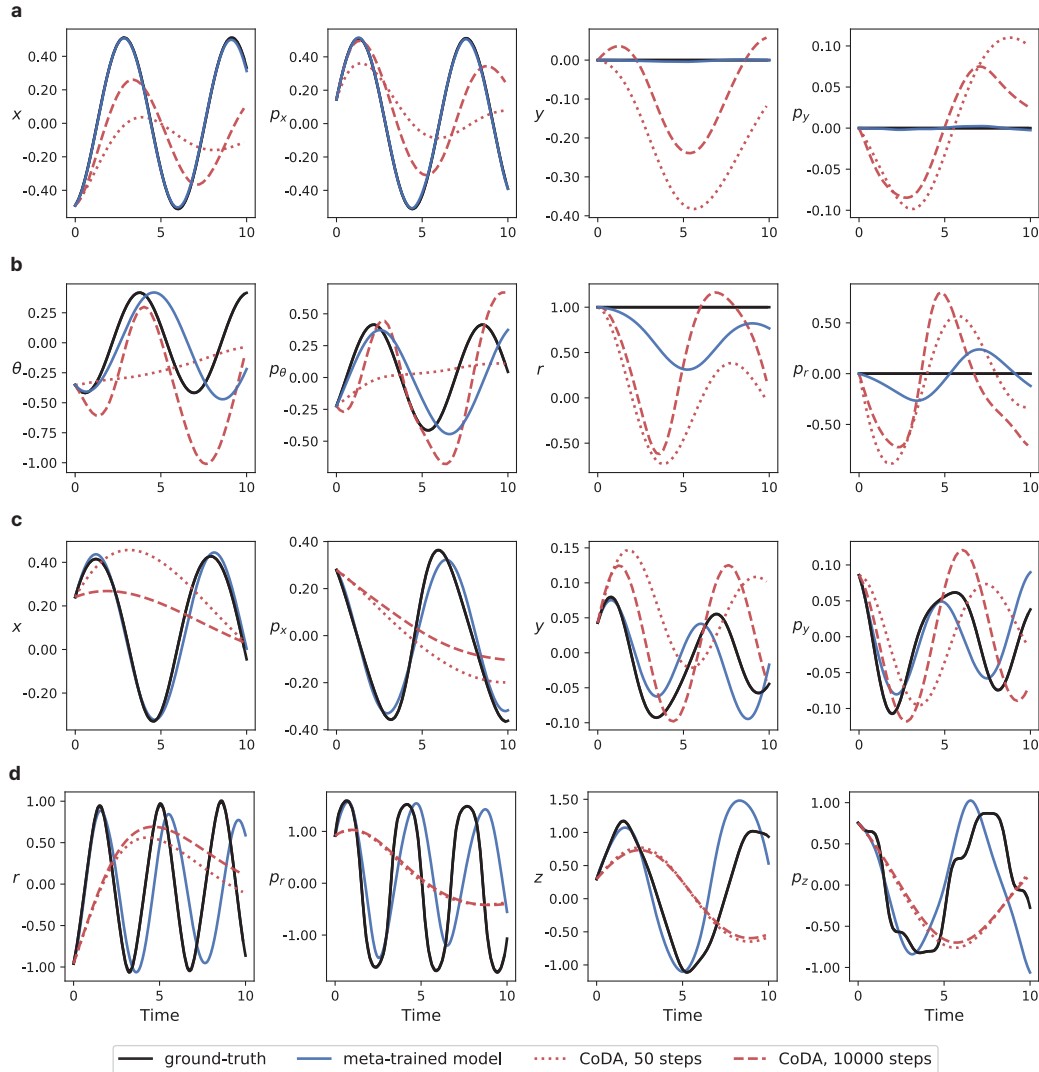

Figure S.6: The prediction of each system dynamics from the meta-trained after 50 adaptation steps, and CoDA after 50 and 10000 steps. Each row (a) to (d) corresponds to the predictions tested with mass-spring, pendulum, Hénon-Heiles, and the magnetic mirror system respectively.

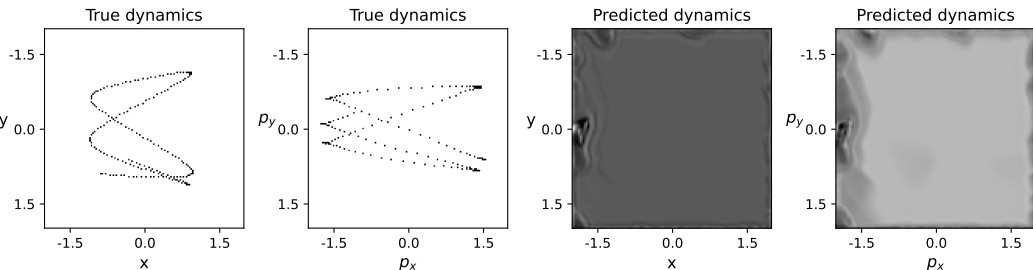

Figure S.7: The predicted dynamics of the magnetic-mirror system using DyAd.

## C.4 COMPARISON WITH CODA

The comparison of the meta-trained model between CoDA and DyAd is depicted in Figure S.6. From the results, the dynamics from CoDA after 50 steps (red dotted line) are not sufficient to match the performance of the meta-trained model, so we also made a comparison with the CoDA after 10000 steps (red dashed line), which also failed to meet the expectations.

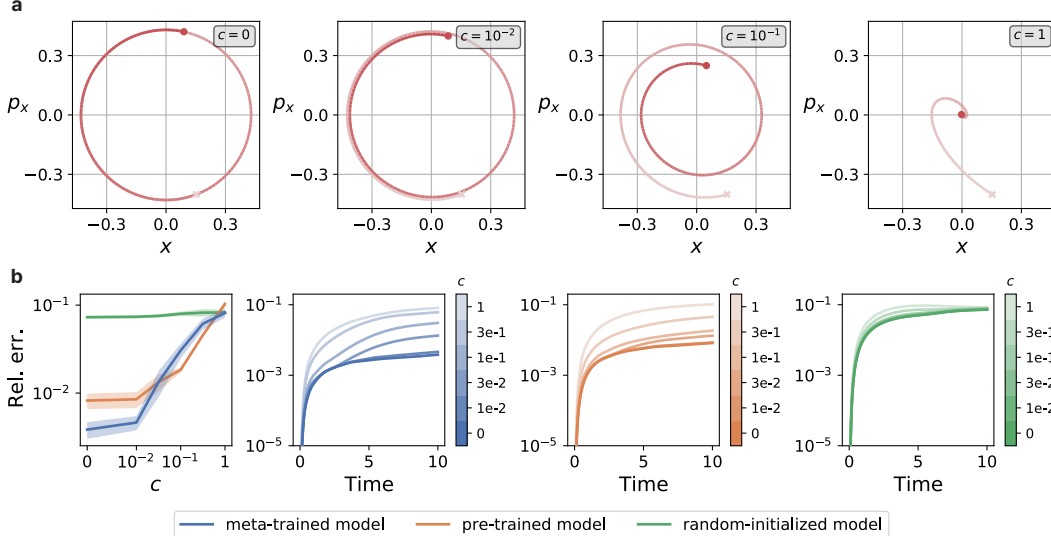

Figure S.8: (a) shows the dissipative dynamics of the mass-spring system as the damping coefficient $c$ changes. The leftmost subfigure in (b) indicates the geometric moving average of the relative error at time 10 for different values of damping coefficient $c$. The three right subfigures in (b) represent the relative error through the time rollout with different values of $c$ for each three models.

Although CoDA is reported to outperform MAML-based methods in the scope of generalization across environments within a fixed system, our results show that MAML-based method advantages CoDA in our broader scope of definition (i.e. generalization across different types of systems).

## C.5 ADAPTATION WITH DYAD

Contrary to CoDA, where inputs are represented by the state variables $(q, p)$ of the system, DyAd relies on image sequence inputs. To this end, we generated sequences of images with dimensions $2 \times 128 \times 128$, where each channel corresponds to the $xy$ space and $p_x p_y$ space, respectively.

However, the weak supervision term in DyAd's encoder loss (the first term in Equation 7),

$$\mathcal{L}_{\text{enc}} = \sum_{c \in \mathcal{C}} \|\hat{c} - c\|^2 + \alpha \sum_{i,j,c} \|\hat{z}_c^{(i)} - \hat{z}_c^{(j)}\|^2 + \beta \sum_{i,c} \|\|\hat{z}_c^{(i)}\|^2 - m\|^2 \tag{7}$$

where $g$ is the encoder network, $x$ is the input, $\hat{z}^{(i)} = g(x^{(i)})$, and $\hat{c}^{(i)} = W\hat{z}_c^{(i)} + b$ is an affine transformation of $z_c$ which act as a hidden feature for task $c$, was not directly applicable to our scenario due to the varying nature of physical parameters across different systems. To adapt this notion to our needs, we have chosen to represent the physical parameter as the energy of the system. While it may not be exact to say that the explicit functional form of energy is common to all systems, the use of energy as a form of weak supervision can effectively utilize the minimal concept of physical parameters across different dynamical systems. We conducted an adaptation task for the magnetic-mirror system using the meta-trained model, which was previously trained on mass-spring, pendulum, and Hénon-Heiles systems. The results show that, as illustrated in Figure S.7, DyAd is not appropriate to our scope of domain generalization.

## C.6 ADAPTATION TO DISSIPATIVE SYSTEM

Here, we performed an adaptation task to the damped mass-spring system with damping term $-c\dot{x}$ with the same meta-trained model that was used to adapt to the original mass-spring system (i.e. meta-trained with conservative pendulum, Hénon-Heiles, magnetic-mirror system). From Figure S.8, we can see that the meta-trained model with conservative systems cannot easily make an adaptation to a dissipative system, which indicates a definite limitation in our problem setting.

### C.7 DEMONSTRATION ON LARGER SYSTEMS

In this section, we present preliminary results for the two-body and three-body tasks. We recognize that the training conditions, such as the learning rate for adaptation, were not sufficiently refined. As a result, while the CKA exhibits behavior similar to that observed in other single-particle tasks as shown in Figure S.10, the prediction accuracy remains suboptimal, as illustrated in Figure S.9. The predicted trajectories are depicted in Figure S.11. It should be noted that while none of the models successfully capture the overall dynamics, the meta-trained model tries to capture the trend than the others. This suggests that with further refinement of the training conditions, the meta-trained model holds promise for achieving more accurate predictions.

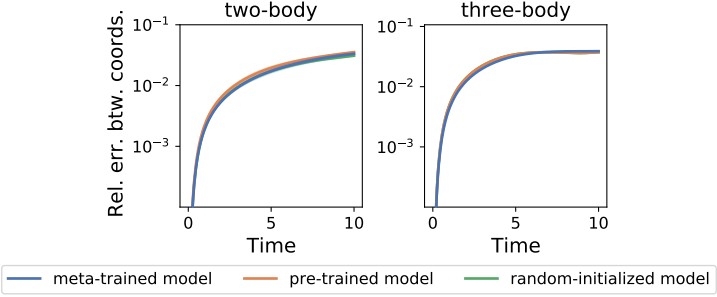

Figure S.9: The relative error throughout time rollout for the meta-trained, pre-trained, and random-initialized model. The solid line and the shaded area each represent the average and the standard error of 10 runs.

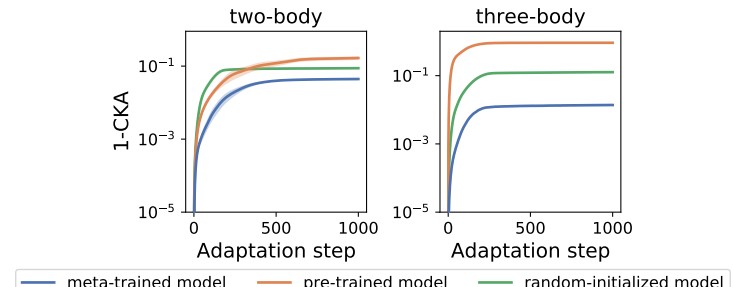

Figure S.10: The measured CKAs in the last layer between the model before making adaptation and during adaptation. The solid line and the shaded area each represent the average and the standard error of 10 runs.

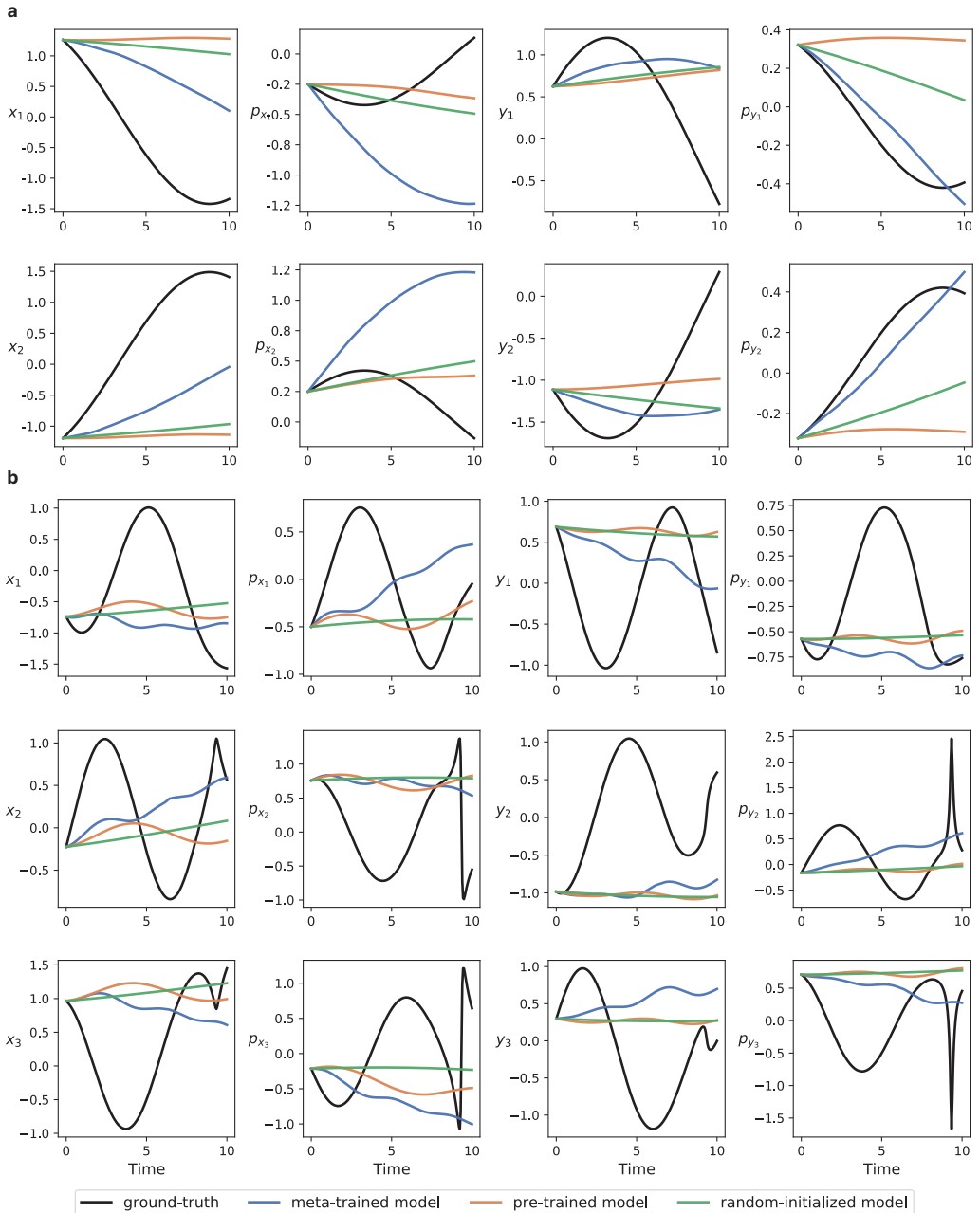

Figure S.11: The example predicted trajectories of (a) two-body, (b) three-body systems.

