# OpenReview forum: "Towards Cross Domain Generalization of Hamiltonian Representation via Meta Learning"
_ICLR.cc/2024/Conference — ICLR 2024 poster_

### Official Review · Reviewer_GtBu · 2023-10-28

**Soundness:** 2 fair
**Presentation:** 3 good
**Contribution:** 2 fair
**Rating:** 5
**Confidence:** 3

**Summary:**

This work leverages the power of meta-learning algorithms coupled with graph neural networks to find a shared representation of physical systems across various functional forms of Hamiltonian. In contrast to previous work, here the focus is on obtaining a representation which is valid across different physical systems. The performance of the framework is evaluated over a range of physical systems aiming to showcase its adaptivity to unseen settings.

**Strengths:**

$\underline{\textrm{Originality}}$: the paper presents great originality in providing a Hamiltonian representation learning framework that can generalize across multiple system domains. This is in contrast to the common approach of providing system-specific models.


$\underline{\textrm{Quality}}$: the paper is very well written. First, it provides the reader with all necessary background as well as motivation for the task addressed. Next, methods and results are well constructed.


$\underline{\textrm{Clarity}}$:  the main ideas conveyed in this paper are clearly constructed and explained and supplementary information assists with providing further details and ablation studies.


$\underline{\textrm{Significance}}$:  the main significance of the paper is in defining a new task, generalizing upon existing approaches, and suggesting to derive a framework that learns a representation that is not domain-specific.

**Weaknesses:**

The paper presents an appealing goal, providing generalized Hamiltonian representations consistent across different physical domains. However, given the presented quantitative and qualitative results it is hard to judge the actual generalization and performance of the framework as detailed in the following points:

1. The notion of $\textit{generalization}$: ideally when discussing generalization in DL we would like to obtain a single pre-trained model which can then be used for diverse applications. With respect to the presented framework, this would suggest training the model(s) on a single task and then using the same network for prediction on all held-out systems. Similar to the setting presented in Ricci et al. (2023). However, here presented results always consider a single held-out-system. Providing an ablation over the number of systems used for training will allow for strengthening the claim of generalization and applicability for real-world applications.

2. Baselines: it would be beneficial to extend the baselines presented in the paper in two directions:
(i) optimal;  Training over the tested task, using all regimes (meta-, pre-, and vanilla HNN). This will allow a better assessment of the quality of the generalized model and (ii) within system generalization; following the background presented in section 2 it will be valuable to add a comparison to frameworks that are similar in nature and allow $\textit{within}$ model generalization, e.g. CoDA (Kirchmeyer et al. 2022) or within the same functional form of the Hamiltonian, e.g. iMODE (Li et al. 2023). Here training over the same train-test splits.

**Questions:**

1. Can the authors provide additional ablation studies following the weaknesses presented above? Specifically, it will be beneficial to present the performance as a function of the number of systems used in training (see weaknesses 1.) and add additional baselines  (see weaknesses 2.)
2. Judging from the presented results the current framework is not suitable for larger systems, could the authors suggest possible extensions that may allow? What would be the necessary refinements that could be incorporated in the meta-learning configuration to allow for that?

---

> ### Author Response · Authors · 2023-11-21
> **Response to Reviewer GtBu**
>
> Thank you for your efforts in providing insightful feedback on our work. We appreciate the reviewer's recognition of the originality of our work compared to other related works. We have thoroughly examined the weaknesses and questions and revised our manuscript as follows (we also recommend the reviewer check the overall response).
>
> **_List of changes in the manuscript_**:
> > 1. Section 4.4 and SM C.3 are revised to add existing baseline results according to Q1
>
> **Q1** Can the authors provide additional ablation studies following the weaknesses presented above? Specifically, it will be beneficial to present the performance as a function of the number of systems used in training (see weaknesses 1.) and add additional baselines (see weaknesses 2.)
>
> > **A1**
> > We appreciate the reviewer for suggesting additional experiments to strengthen our paper.
> > Regarding weakness 1, it's important to clarify that our primary objective was to explore the generalization of Hamiltonian dynamics for predicting a single, unknown held-out system, underpinned by ample data. While the experiment proposed by the reviewer is indeed beneficial, it aligns more closely with further challenging scenarios involving adaptation to multiple unknown systems. Such situations extend beyond the current scope of our research, which focuses on a singular system context.
> > For weakness 2, we originally thought that it was hard to directly compare our method with other prior baselines as the definition of the scope of generalization is different. However, we agree that adding such baselines would be beneficial. As such, we choose CoDA ([Kerchmeyer et al. 2022](https://arxiv.org/abs/2202.01889)) as a representative method for domain generalization in learning the physical dynamics (as it is reported to be superior among other related methods according to the CoDA authors). As expected, CoDA failed to adapt to an unseen system with the same number of steps, even for sufficient steps up to 10000. We have revised Section 4.4 and SM C.3 of our manuscript with the CoDA baseline results and included the corresponding discussion.
>
>
> **Q2** Judging from the presented results the current framework is not suitable for larger systems, could the authors suggest possible extensions that may allow? What would be the necessary refinements that could be incorporated in the meta-learning configuration to allow for that?
>
> > **A2**
> > Although the results for the large systems are quite poor in the reported results, we want to mention that our approach can handle the problem when the degree of freedom of the prepared data is not consistent throughout the system, noting that the previous methods discussed in Section 2.2 and 2.3 cannot be used in such scenarios. Although we cannot give the exact refinements in our preliminary results, possible ways could be 1) tune the update method in meta-learning, e.g. [Lee et al. 2021](https://arxiv.org/abs/2102.11544) report that using ANIL variation gives more accuracy compared to vanilla MAML, and 2) strictly search the hyperparameter space.

---

> > ### Comment · Reviewer_GtBu · 2023-11-21
> >
> > Thank you for the detailed response; while some of my concerns have been addressed others remain hence i choose to keep my score (mainly, while not set as a primary objective, i believe the _generalization_ notion stands as a big weakness).

---

> ### Author Response · Authors · 2023-11-23
>
> We acknowledge the reviewer's concern regarding the generalization aspect of our study. The absence of the suggested ablation experiment in our work stemmed from our assessment that the number of systems (1 to 3) required for such an experiment would not be adequate for the adaptation task at hand. Instead, to evaluate such generalization, we directly tested the meta-trained model's performance on multiple held-out systems. Following the reviewer's suggestions, we extended our analysis to include the 2D harmonic oscillator system, utilizing the same meta/pre-trained models from our initial experiments. This approach aimed to assess the model's capability in generalizing across multiple held-out systems. The results showed that the meta-trained model significantly outperformed the baselines in terms of both coordinate prediction and energy estimation, thus reinforcing its generalization ability on a broader range of systems. We have updated Section 4.4 and SM C.3 in our manuscript to incorporate these findings and the corresponding discussions. We hope that the additional response will address the concerns raised and appreciate the opportunity to clarify these aspects within the limited discussion period.

---

> > ### Author Response · Authors · 2023-11-23
> >
> > Previously, we have only added 2D harmonic oscillator system as a new example to check the adaptation to multi held-out systems. To give more concreteness to the point of the **notion of generalization** that the reviewer has given to us, we also added Kepler system as a new system to test. Accordingly, we have revised Section 4.4 and SM C.3. We deeply appreciate the consideration given to our responses, especially within the limited discussion period.

---

### Official Review · Reviewer_xLig · 2023-10-31

**Soundness:** 3 good
**Presentation:** 3 good
**Contribution:** 3 good
**Rating:** 6
**Confidence:** 2

**Summary:**

This paper proposed a novel meta-learning method aiming to learn a unified Hamiltonian representation such that it can be generalized to unseen physical systems. Hamiltonian Neural Networks were utilized as the backbone of the method for learning the unified Hamiltonian representations of different physical systems, via meta-learning pipelines of a variation of MAML. Experiments demonstrated the proposed methods achieved lower relative error of trajectories and energy when adapted to different systems.

**Strengths:**

(1) Unlike many existing works that focus on learning system dynamics under similar physical law, the proposed methods aim to learn the unified representations across diverse system domains via meta-learning the Hamiltonian of the given system. This sounds significant and promising.

(2) Both quantitative and qualitative results demonstrated the proposed method achieved better adaptation to various new systems compared with baselines.

**Weaknesses:**

The evaluation can be strengthened by considering comparing the proposed methods with other Few-shot Learning and Physics-informed Neural Networks methods for system domain generalization under both "consistent" and "different" physical laws.

**Questions:**

In Figure 4, why does a lower CKA value of the meta-trained model suggest it learned more similar representations during adaptation? As the authors mentioned, should a low CKA value indicate different representations?

---

> ### Author Response · Authors · 2023-11-21
> **Response to Reviewer xLig**
>
> We appreciate your thoughtful effort in reviewing this work. We are delighted to hear that the reviewer has recognized the contribution of our work in contrast to many existing works. We have carefully considered the points raised by the reviewer and have revised our manuscript as follows (we also recommend that the reviewer check the overall response).
>
> **_List of changes in the manuscript_**:
> > 1. Section 4.4 and SM C.3 are revised to add existing baseline results according to W1
>
> **W1** The evaluation can be strengthened by considering comparing the proposed methods with other Few-shot Learning and Physics-informed Neural Networks methods for system domain generalization under both "consistent" and "different" physical laws.
>
> > **A1**
> > We first thought that it would be difficult to directly compare our method with other previous baselines, as the definition of the scope of generalization is different. However, we strongly agree with the reviewer that our evaluation can be strengthened by comparing the proposed approach with other existing methods. We added a baseline experiment with CoDA ([Kerchmeyer et al. 2022](https://arxiv.org/abs/2202.01889)) which is considered a representative method for domain generalization in learning physical dynamics (as it is reported to be superior to other related methods according to the CoDA authors). As expected, CoDA failed to adapt to an unseen system with the same number of steps, even for sufficient steps up to 10000. We have revised section 4.4 of our manuscript with the CoDA baseline results and discussion.
>
> **Q1** In Figure 4, why does a lower CKA value of the meta-trained model suggest it learned more similar representations during adaptation? As the authors mentioned, should a low CKA value indicate different representations?
>
> > **A1**
> > We acknowledge that our label notation in Figure 4 could be quite misleading. The y-axis represents the 1-CKA (1 minus CKA) value, not the raw CKA value. Therefore, the lower y-axis in Figure 4 indicates a high CKA value, which leads to more similar representations during adaptation.

---

> > ### Comment · Reviewer_xLig · 2023-11-22
> >
> > Thank the authors for their clarification and the additional experiments. I will maintain my score.

---

> > > ### Author Response · Authors · 2023-11-23
> > >
> > > Although there were some additional results were presented in a rush within the limited discussion period (including additional DyAd baseline results), we hope that these points can resolve the remaining concerns. Once again, thank you for your effort in reviewing our work.

---

### Official Review · Reviewer_jozM · 2023-10-31

**Soundness:** 2 fair
**Presentation:** 3 good
**Contribution:** 3 good
**Rating:** 6
**Confidence:** 3

**Summary:**

The paper reports the performance of MAML applied to domain generalization over different Hamiltonian dynamics. The performance gain by MAML is analyzed with multiple indices over different combinations of meta-training and meta-test data. The key findings include the superiority of the meta-trained models compared to the pre-trained and randomly initialized models and an implication that by meta-learning, the representations obtained by the models tend to be more specific to each system.

**Strengths:**

The experiments clearly show the superiority of meta-learning, at least within the limited number of Hamiltonian systems.

The paper is well written. The motivation, the method, and the experimental results are very clearly reported.

**Weaknesses:**

The limitation of meta-learning for (Hamiltonian) dynamics is not clearly investigated. This makes it difficult to assess the range where the claims made in the paper should be valid. In other words, the claims are rather weak because their applicability seems unbounded with the current set of experiments. When the meta-learning approaches for dynamics may not be beneficial? For example, what happens if you meta-train a model only with conservative systems and try to adapt it to a dissipative system? Such experiments to investigate the limitations of the empirical findings would strengthen the paper.

The paper only reports the performance of a well-known method (MAML) merely applied to a particular setting. This could certainly be a kind of contribution in which ICLR audience may be interested, but I think that in such a paper, with a purely experimental point of view, the claims should be made more carefully. Specifically, as stated above, the cases where meta-learning is not necessarily beneficial should also be revealed, with which the claims would become more falsifiable and convincing.

**Questions:**

I don't have particular questions. It would be great if the authors could additionally report the results of some experiments to investigate the limitation of meta-learning in this context, although this is not a question.

---

> ### Author Response · Authors · 2023-11-21
> **Response to Reviewer jozM**
>
> Thank you for your efforts in providing valuable comments on our work. We appreciate the reviewer's comment about the points that should be further considered from the aspect of our experimental point of view. We have thoroughly considered the concerns raised by the reviewer and revised our manuscript as follows (we also recommend the reviewer check the global response).
>
>
> **_List of changes in the manuscript_**:
> > 1. Section 4.5 and SM C.4 are revised to add additional experiments according to W1
>
> **W1** I don't have particular questions. It would be great if the authors could additionally report the results of some experiments to investigate the limitation of meta-learning in this context, although this is not a question.
>
> > **A1**
> > It should be noted that, in an attempt to provide further insight beyond the experimental results for the generalized model, we have provided the CKA analysis of the learned representations for each of the models. This was also the reason why we only compared the meta-trained model with the pre-trained and randomly initialized model, and not with the existing baselines in the first place. Nevertheless, we strongly agree with the reviewer that our results and claims should be made more carefully from our paper’s experimental point of view, and we appreciate the suggestion to make our claims more thorough by exploring the limitations of meta-learning for learning a unified representation of Hamiltonian dynamics.
> Consequently, we consider the example suggested by the reviewer (meta-training the model with conservative systems and adapting it to a dissipative system), which we believe is a good way to explore this. As expected, the meta-trained model (as well as the pre-trained and randomly initialized model) failed to adapt to a dissipative system, showing a clear limitation of our approach. We discuss the reason why, for the case of a dissipative system, Hamilton's equation (Equation 1 in the manuscript) does not hold, and thus the HNN loss (Equation 2 in the manuscript) is not appropriate for dissipative systems. We argue that to make use of our method, both the system in the data distribution for meta-training and adaptation should share the same nature so that both satisfy the given loss of the implementation. We have added the corresponding experimental results and discussions in Section 4.5 and SM C.4.

---

> > ### Comment · Reviewer_jozM · 2023-11-21
> >
> > Thanks for the response. The revised manuscript is clearer in terms of the specific contribution, and the limitation shown in the new experiments is convincing. I modified my evaluation accordingly.

---

> > > ### Author Response · Authors · 2023-11-23
> > >
> > > We are glad to hear that the concerns from the reviewer were relieved with the additional experiments that the reviewer has suggested. Once again, we greatly appreciate your effort in reviewing our work.

---

### Official Review · Reviewer_hDu2 · 2023-10-31

**Soundness:** 3 good
**Presentation:** 3 good
**Contribution:** 3 good
**Rating:** 6
**Confidence:** 4

**Summary:**

This paper proposes to use meta-learning to learn generalized representations across different types of dynamical systems. The meta-learning step helps improve the adaptation to unknown systems with fewer data points (compared to randomly initialized and pre-trained baselines) by virtue of generalized representations. The authors also analyze the representations learned by different baselines and meta-learning by using centered kernel alignment (CKA) to gain insights into better performance by the meta-learning model.

**Strengths:**

- The paper is easy to follow and the motivation to learn a generalized model is clear.
- The experiments are performed with different numbers of data-points for the adaptation task to evaluate the robustness of the approach.
- The analysis using CKA gives further insight into how the meta-learning model learns closer representations of the adaptation task.
- The implementation and task curation are described in detail for reproducibility.

**Weaknesses:**

- The main contribution of the paper seems to be utilizing meta-learning to efficiently adapt to new systems. However, it is not clear from the paper if it is as simple as just using off-the-shelf implementation or if there are some challenges to doing this.
- Also, I would like to see some discussion around why meta-learning is preferred over other representation learning methods e.g. Domain Generalization, and why optimization-based methods surpass other approaches in meta-learning.
- The experiments are not sufficient. First, there is no comparison with existing baseline models in domain generalization or meta-learning. Second, the comparison of the meta-model and pre-trained model is not fair, and I would suggest the author fine-tune the pre-trained model on the K-shot support set. Third, no visual comparison of the predicted dynamics and the ground truth, making the conclusion less convincing.

**Questions:**

Please check the weaknesses above.

---

> ### Author Response · Authors · 2023-11-21
> **Response to Reviewer hDu2**
>
> We sincerely appreciate your efforts in reviewing our paper. We are pleased that the reviewer noticed the strengths regarding the CKA analysis of the learned representations. We have carefully reviewed the weaknesses pointed out by the reviewer and have revised our manuscript as follows (we also recommend the reviewer review the overall response).
>
> **_List of changes in the manuscript_**:
> > 1. Section 2.3 is revised to add clarity to the paper based on W1, and W2.
> > 2. Section 4.4 and SM C.3 are revised to add existing baseline results following W3.
> > 3. The titles of Sections 4.1, and 4.2 and the caption of Fig. 3 are updated based on W3.
>
> **W1** The main contribution of the paper seems to be utilizing meta-learning to efficiently adapt to new systems. However, it is not clear from the paper if it is as simple as just using off-the-shelf implementation or if there are some challenges to doing this.
>
> **W2** Also, I would like to see some discussion around why meta-learning is preferred over other representation learning methods e.g. Domain Generalization, and why optimization-based methods surpass other approaches in meta-learning.
>
> > **A1, A2**
> > We presume that weaknesses 1 and 2 arise from the lack of clarity that we have written in the paper. Existing previous methods have clear limitations; 1) they do not guarantee generalization across diverse systems, and 2) to do so, they need to be flexible with respect to the varying degrees of freedom of the systems. Our approach aims at resolving these challenges by jointly using GNNs and meta-learning. Accordingly, we have revised section 2.3 to clearly state the points mentioned by the reviewer in Q1 and Q2.
>
> **W3** The experiments are not sufficient. First, there is no comparison with existing baseline models in domain generalization or meta-learning. Second, the comparison of the meta-model and pre-trained model is not fair, and I would suggest the author fine-tune the pre-trained model on the K-shot support set. Third, no visual comparison of the predicted dynamics and the ground truth, making the conclusion less convincing.
>
> > **A3**
> > - First, we acknowledge the insufficient comparison with existing baselines. Originally, we considered it to be hard to directly compare our method with other prior baselines due to the different definitions of the scope of generalization, we added a baseline experiment with CoDA ([Kerchmeyer et al. 2022](https://arxiv.org/abs/2202.01889)) which is considered a representative method for domain generalization in learning physical dynamics (as it is reported to be superior among other related methods according to the CoDA authors). As expected, CoDA failed to adapt to an unseen system with the same number of steps, even for sufficient steps up to 10000. We have revised Section 4.4 and SM C.3 of our manuscript with the CoDA baseline results and discussion.
> > - Second, while we appreciate the reviewer's perspective, our work is at the cornerstone in our scope of the problem definition, and thus to show both the possibility and the effectiveness of the unified representation of Hamiltonian dynamics, we chose as baselines the 1) pre-trained model that used the same total number of gradient steps during meta-training, and the 2) randomly initialized model, following the approach of the prior work of [Finn et al. 2017](https://arxiv.org/abs/1703.03400) and [Lee et al. 2021](https://arxiv.org/abs/2102.11544).
> > - Third, we respectfully ask the reviewer to check Section 4.2, where the dynamics of the predicted systems are discussed using Figure 3. However, we acknowledge that the presentation of the predicted dynamics written in the paper should be more explicit and accordingly we have updated the title of Sections 4.1, 4.2 and the caption of Figure 3.

---

> > ### Comment · Reviewer_hDu2 · 2023-11-21
> >
> > Thank you for your detailed response. Although some of my concerns have been addressed, there are still major questions remaining to be answered:
> > 1. The description of the methodology is not clear. Both the proposed work and CoDA are based on optimization-based meta-learning and one of the key differences is the use of graphs. However, the details of what the vertices and edges in the graph represent and how the graph is constructed are missing. Also, it would be good to have an ablation study on the proposed method without graph representation.
> > 2. There are lots of non-MAML-based meta-learning models working on physics dynamics and they deserve to be compared. For instance, DyAd [1] is a model-based meta-learning with weak supervision for turbulence flow on heterogeneous domains; Sequential Neural Processes [2] has a similar see,tting of generalizing on sequential dynamics; meta-SLVM [3] addresses the adaptation across dynamics via Bayesian meta-learning and has experimented on different physics model. These works should be mentioned as related works and compared to demonstrate how the proposed work improves the generalization across different dynamics.
> >
> > I choose to keep my score unless the concerns above are addressed.
> >
> > [1] Wang et al, Meta-learning dynamics forecasting using task inference, 2022
> > [2] Singh et al, Sequential neural processes, 2019
> > [3] Jiang et al, Sequential Latent Variable Models for Few-Shot High-Dimensional Time-Series Forecasting 2023

---

> ### Author Response · Authors · 2023-11-23
>
> We acknowledge your detailed comment regarding our response. Below we provide a response for the reviewer’s additional comments.
>
> 1. As the reviewer pointed out, one of the key differences between our proposed work and CoDA is the usage of graphs. We would like to add one more major difference as follows. According to the CoDA paper, CoDA is based on the assumption that the *environment* $e \in \mathcal{E}$ is from a specific dynamics $f$ (where $\frac{dx(t)}{dt} = f(x(t))$) such that the functional form of the dynamics set from the training data and the dynamics set of the unseen data should be the same. In other words, the context of environment-specific parameters is the same across the environments. This way of formulation can also be found for DyAd [1], and meta-SLVM [3]. However, in our setting, we do not impose such assumptions such that the term *environment* (which was a generalization within a fixed dynamics form) can be extended in the context of the term *domain* (now a generalization across different dynamics) we used in our manuscript.
>     And for the missing details about how the graphs represent the system, we followed the settings from [Sanchez-Gonzalez et al. 2019](https://arxiv.org/abs/1909.12790), and [Bishonoi et al. 2023](https://openreview.net/forum?id=Ugl-B_at5n) such that the nodes represent the state $(\vec{q}, \vec{p})$, while the edge features are not used for now. We revised Section 3.2 accordingly.
> 2. We acknowledge the reviewer's observation regarding potential misunderstandings on our part. We focused on reporting baseline results for DyAd [1] within a specific system context. Our additional evaluation included testing the magnetic-mirror system adaptation task using a meta-trained model, which encompassed systems of mass-spring, pendulum, and Hénon-Heiles. Notably, DyAd showed limited adaptation capabilities in this context, struggling significantly with the magnetic-mirror system.
>     This observation leads us to a critical point: methods such as CoDA, DyAd, and meta-SLVM, which aim to generalize within a fixed functional form of dynamics, may not be directly applicable to our concept of domain generalization (as we have previously discussed in the above-numbered point 1.). To align DyAd more closely with our generalization scope, we made a subtle modification to the parameter $c$ in the encoder loss's first term (weak supervision term), representing it as the Hamiltonian of the system. While it may be hard to argue that the explicit form of Hamiltonian is not shared across the system, we viewed this as a minimal form of weak supervision, leveraging the universal concept of energy in dynamical systems.
>     And for the meta-SLVM, although it adapted well to several physical scenarios (namely bouncing balls under 4 gravity, pendulums, and mass springs with each four different physical constants), their adaptation was tested under the same type of physics; among the four dynamics for each physical scenarios, three were used for meta training, and the remaining one was used for meta testing.
>     Therefore, despite the inherent limitations of DyAd and similar methods in this context, we posit that our approach, leveraging a vanilla MAML-based method, demonstrates superior adaptability and effectiveness compared to these baselines. This distinction underscores the rationale behind our methodological choices and the results we observed. Accordingly, we revised Section 2.3 with the discussions and Section 4.4 and SM C.5 with DyAd results. We hope that the additional response will address the concerns raised and appreciate the opportunity to clarify these aspects within the limited discussion period.

---

> > ### Comment · Reviewer_hDu2 · 2023-11-23
> >
> > Thank the authors for the clarification and additional experiments. Below are some suggestions that may improve the presentation of the work based on the discussion above:
> > 1. The authors could highlight the method is able to process unseen domains and add the details of the graph to the main text. The comparison of the proposed work and existing meta-learning works, including the non-MAML-based works, could be discussed in the related works.
> > 2. I appreciate the effort of adding experiments in such limited discussion time, although I still encourage the authors to make more effort on the non-MAML-based works, including the reasons of their inability to adapt to unseen domains and whether the proposed models can be applied to these frameworks.
> >
> > I modified my evaluation accordingly.

---

> > > ### Author Response · Authors · 2023-11-23
> > >
> > > We thank the reviewer for considering the additional baseline results and discussions. With the additional comments that the reviewer has given to us, we will revise our manuscript including the highlight that our method is able to process to unseen domains and so on. Once again, we heartily thank the reviewer for the effort in reviewing our work.

---

### Author Response · Authors · 2023-11-21
**Overall response**

We would like to thank all of the reviewers for their constructive and valuable feedback on our work, and furthermore recognizing the strengths of our paper as follows;

- the originality of our work in contrast to previous studies; learning the unified representation across diverse Hamiltonian system domains (**Reviewer xLig** and **Reviewer GtBu**)
- the importance of the experimental aspects of our work; analysis of the unified representation of Hamiltonian dynamics using CKA (**Reviewer hDu2 and Reviewer jozM**)


We rigorously addressed and incorporated the feedback to strengthen our work. Below are the main points of the revised manuscript (the changes in the manuscript are indicated as **bold**).


1. Most reviewers suggested a comparison with the baseline of the existing methods described in sections 2.2, and 2.3. The reason we did not include the previous generalization methods was that it was rather difficult to make a direct comparison with our model. Our model aims to learn a unified representation of the Hamiltonian dynamics across the types of physical systems, while the previous studies focus on the scenario of generalizing the dynamics across multiple environments within a fixed system situation. However, we agree with the reviewers that adding the existing baselines would give more concreteness to our result, and added the CoDA ([Kerchmeyer et al. 2022](https://arxiv.org/abs/2202.01889)) (which showed a significantly high performance among various generalization methods across environments within a system, according to the CoDA authors) baseline experiment. **We updated Section 4.4 and SM C.3 with the CoDA baseline and again verified the originality and superiority of our approach.**
2. We have added experiment to further discuss the limitation of our method, which is indeed a crucial point to make an argument from our experimental point of view. The added experiment is an adaptation task to the dissipative mass-spring system with the meta-trained model with conservative other systems. The meta-trained model fails to adapt to the dissipative system, indicating a possible limitation of our approach. **We updated Section 4.5 and SM C.4 with the corresponding additional experiments and discussions to explore the limitations of our work.**
3. Although all reviewers commented that the motivation and presentation of our paper are clearly written, there were some missing points regarding the implementation challenges solved. Although the existing generalization methods (as noted in the first numbered point above) propose new frameworks to learn the generalized dynamics across environments, their applications are limited when we want to generalize across the system type itself, and especially when the degree of freedom of the system changes in the data distribution. **We updated Section 2.3 to clearly state that our approach straightened out the challenges.**
4. Some of the text, such as section titles and figure captions, has been updated to reflect the comments made by reviewers. **We fixed the title of Section 2.2, 2.3, the caption of Figure 3, and moved Table 1. to SM A.3**

Our manuscript has been revised accordingly (blue; major updates, red; minor updates such as typos) and we kindly ask all reviewers to check the updated part of the paper. We hope that our revised manuscript and responses will have a positive impact on the final evaluation.
Once again, we thank all reviewers for their efforts in this review and look forward to further discussions.

---

> ### Author Response · Authors · 2023-11-23
>
> Dear reviewers, we would like to inform you that several additional experiments are updated in our manuscript as follows.
>
> **_Updates_**:
> > 1. DyAd ([Wang et al. 2022](https://arxiv.org/abs/2102.10271)) baseline results tested on magnetic-mirror system are added in Section 4.4 and SM C.5.
> > 2. To further emphasize the generalization ability of multi held-out systems, the adaptation task to a new 2D harmonic oscillator system is added in Section 4.4 and SM C.3.
> > 3. Section 2.2 has been revised to provide a more discussion of methods of generalizing dynamics based on meta-learning. This revision outlines the inherent limitations of these methods and describe how our approach differs.
>
> We are grateful for the opportunity to provide further clarification and deeply appreciate the consideration given to our responses, especially within the limited discussion period. We hope that our additional responses will have a positive impact on the final evaluation.

---

> > ### Author Response · Authors · 2023-11-23
> >
> > Dear reviewers, we would like to inform you that the Section 4.4 and SM C.3 are updated with an additional test system; Kepler system, to add more concreteness to the ability of generalization of multi held-out systems.
> >
> > Once again, we deeply appreciate the consideration given to our responses, especially within the limited discussion period.

---

### Meta-Review · Area_Chair_9tyn · 2023-12-05

**Metareview:**

This paper introduces an approach for meta-learning across a variety of cross-domain Hamiltonian systems. The reviewers generally found the paper well-written and well-motivated and the experiments interesting and compelling within the setting of Hamiltonian systems. The discussion highlighted some weaknesses of the paper, particularly related to experimental baselines and limitations. The author response and updated manuscript addressed many of the most significant weaknesses, and as such I recommend acceptance.

**Justification For Why Not Higher Score:**

The limitations of the approach and setting are not clearly defined or investigated, and therefore the scope of applicability and broader appeal remains unclear.

**Justification For Why Not Lower Score:**

The paper is well-written and interesting, the approach sensible, and the qualitative and quantitative results are strong and the benefit of meta-learning appears clear.

---

### Decision · Program_Chairs · 2024-01-16

Accept (poster)